# Metronomic Chemo-Endocrine Therapy (FulVEC) as a Salvage Treatment for Patients with Advanced, Treatment-Refractory ER+/HER2-Breast Cancer—A Retrospective Analysis of Consecutive Patients Data

**DOI:** 10.3390/jcm12041350

**Published:** 2023-02-08

**Authors:** Anna Buda-Nowak, Łukasz Kwinta, Paweł Potocki, Anna Michałowska-Kaczmarczyk, Agnieszka Słowik, Kamil Konopka, Joanna Streb, Maciej Koniewski, Piotr J. Wysocki

**Affiliations:** 1Department of Oncology, Jagiellonian University Medical College, University Hospital, 30-501 Krakow, Poland; 2Department of Oncology, Jagiellonian University Medical College, 31-008 Krakow, Poland; 3Institute of Sociology, Jagiellonian University, 30-962 Krakow, Poland

**Keywords:** breast cancer, metronomic chemotherapy, chemo-endocrine therapy, FulVEC

## Abstract

Background: Breast cancer, with 2.3 million new cases and 0.7 million deaths every year, represents a great medical challenge worldwide. These numbers confirm that approx. 30% of BC patients will develop an incurable disease requiring life-long, palliative systemic treatment. Endocrine treatment and chemotherapy administered in a sequential fashion are the basic treatment options in advanced ER+/HER2- BC, which is the most common BC type. The palliative, long-term treatment of advanced BC should not only be highly active but also minimally toxic to allow long-term survival with the optimal quality of life. A combination of metronomic chemotherapy (MC) with endocrine treatment (ET) in patients who failed earlier lines of ET represents an interesting and promising option. Methods: The methodology includes retrospective data analyses of pretreated, metastatic ER+/HER2- BC (mBC) patients who were treated with the FulVEC regimen combining fulvestrant and MC (cyclophosphamide, vinorelbine, and capecitabine). Results: Thirty-nine previously treated (median 2 lines 1–9) mBC patients received FulVEC. The median PFS and OS were 8.4 and 21.5 months, respectively. Biochemical responses (CA-15.3 serum marker decline ≥50%) were observed in 48.7%, and any increase in CA-15.3 was observed in 23.1% of patients. The activity of FulVEC was independent of previous treatments with fulvestrant of cytotoxic components of the FulVEC regimen. The treatment was safe and well tolerated. Conclusions: Metronomic chemo-endocrine therapy with FulVEC regimen represents an interesting option and compares favorably with other approaches in patients’ refractory to endocrine treatments. A phase II randomized trial is warranted.

## 1. Introduction

Breast cancer (BC) is the most commonly diagnosed cancer type, accounting for one in eight cancer diagnoses worldwide. Over the last 30 years, the incidence of BC has continuously increased, while mortality demonstrated an opposite trend. There were about 2.3 million new breast cancer cases and 0.7 million breast cancer deaths worldwide in 2020. Breast cancer is the second most common cause of death from cancer in women in the United States and numbers fifth worldwide [1]. There is no doubt, that the majority of BC-related death is observed in patients with disseminated diseases. While only 5% of BC patients are initially diagnosed with metastatic disease, approx. 30% of BC survivors will ultimately metastasize [2]. Over the recent two decades, despite the introduction of many novel targeted therapies for mBC, the prognosis of advanced breast cancer patients is poor, and mBC remains an incurable, fatal condition.

Most BCs express estrogen receptors (ER), which act as a therapeutic target for endocrine therapies representing the central core of the initial systemic treatment in the majority of advanced BC ER+/HER2- patients [3]. The activity of endocrine therapy (ET), even when combined with novel targeted agents such as CDK4/6 inhibitors, PIK3CA, or mTOR inhibitors, is time-restricted, and ultimately, all metastatic BC patients treated with ET will progress. Globally, chemotherapy (CT) is the treatment of choice in ER+/HER2- patients resistant to ET [4]. Although novel approaches based on antibody-drug conjugates are emerging, their timely implementation in clinical practice may be limited by regulatory and financial issues. Most advanced, ET-resistant BCs are treated with single-drug chemotherapy regimens administered intravenously or orally on a long-term (until progression) basis. However, the heterogeneity of advanced BC represents a tremendous clinical challenge in that various metastatic lesions may develop distinct resistance mechanisms to ET and CT, thus making them susceptible to different therapeutic agents [5]. Therefore, combinatory approaches represent a promising treatment strategy for the better systemic control of the disseminated disease. While intravenous multi-drug regimens are usually highly toxic, the metronomic, orally administered polychemotherapy is much better tolerated, and due to its multidirectional activity (anti-proliferative, antiangiogenic, and immunomodulatory), it may provide superior clinical benefits in many tumor types, including breast cancer [6,7,8].

The combination of ET and CT is still controversial based on the non-conclusive evidence from numerous but old clinical studies. However, considering the highly positive data on the combination of ET with antiproliferative agents (CDK4/6i) and many biological similarities between CDK4/6i and metronomic chemotherapy, the combination of ET with CT deserves attention, especially in low- and middle-income countries for which many novel drugs are simply unaffordable [9].

This manuscript presents the results of a retrospective analysis that evaluated the safety and efficacy of a metronomic chemo-endocrine regimen (FulVEC) in advanced, pretreated ER+/HER2- patients.

## 2. Materials and Methods

### 2.1. Patients

We retrospectively collected data on consecutive, advanced BC patients treated with metronomic chemo-endocrine therapy (FulVEC) at Jagiellonian University-Medical College Hospital in Cracow between 2018 and 2022. Metronomic chemotherapy has been initially offered to pretreated, progressing (radiographic and biochemical progression) patients who were deemed unfit or refused further lines of standard intravenous chemotherapy regimens. Furthermore, the treatment was also offered to pretreated, progressing patients who were asymptomatic or mildly symptomatic and were reluctant toward the immediate initiation of intensive, intravenous chemotherapy. Eligible patients had histologically proven advanced (metastatic or locally recurrent) inoperable breast cancer and presented with a performance status of ECOG 0–2. The study has been approved by the local bioethical committee at Jagiellonian University.

### 2.2. Treatment

The metronomic chemo-endocrine therapy FulVEC consisted of fulvestrant 500 mg i.m. administered on days 1, 14, 28, and q1m thereof combined with metronomic, continuous polychemotherapy VEC (vinorelbine 40 mg three times a week, cyclophosphamide 50 mg p.o. qd, capecitabine 500 mg p.o. tid). Dose adjustments were performed based on the evaluation of treatment-emergent adverse events, and particular drug-related AE has led to a stepwise reduction in a given drug. Upon the occurrence of vinorelbine-induced AE, the dosage was reduced from 50 mg tiw to 30 mg q2d. Capecitabine-related AE led to a stepwise reduction from 500 mg tid to 500 mg bid and then to 500 mg qd. The single-step reduction in cyclophosphamide was based on decreased frequencies from 50 mg qd to 50 mg q2d. In the case of non-sufficient dose reduction, the treatment with a particular drug was withheld until the resolution of particular AEs.

Data on the following background characteristics of the patients were collected using standardized data collection instruments: age; ECOG performance status; clinical symptoms; serum tumor markers, including CA-15.3; tumor stage (locally advanced or metastatic); sites of distant metastases; pathological diagnosis including immunohistochemistry.

### 2.3. Analysis of Treatment Efficacy

Since the metronomic chemo-endocrine treatment was initially used as a last-resort therapy for heavily pretreated ER+ BC patients, an objective evaluation of tumor response by imaging modalities was not considered a critical aspect and a justified option in most treated individuals. Clinical decisions were based on the patient’s performance status, disease-related symptoms intensity, and lab results involving blood morphology and the biochemical evaluation of organ functions. Treatment efficacy was analyzed retrospectively by the evaluation of overall survival (OS), progression-free survival (PFS), and treatment-induced changes in the CA-15.3 level. Biochemical response (bRR) was defined as ≥50% serum marker reduction, biochemical stabilization (bSD) was defined as a ≤50% reduction, and biochemical progression (bPD) was defined as any increase in CA-15.3 levels. The biochemical benefit (bBR) was defined as any decline in CA-15.3 concentrations.

### 2.4. Safety Analysis

Data on treatment-related myelotoxicity was obtained by the automatic analysis of laboratory results of the blood samples collected every four weeks during treatment with the FulVEC regimen. Data on other bone-marrow-unrelated AEs and on FulVEC dosage modifications or interruptions were derived directly from patients’ medical history.

### 2.5. Statistical Considerations

Distributions of quantitative variables were summarized with mean, standard deviation, median and interquartile range, whereas distributions of qualitative variables were summarized with the number and percent of occurrence for each of their values.

Hazard ratios of death and progression were estimated for each predictor at interest in a univariate Cox proportional hazard model with Jackknife standard errors. Kaplan-Meier curves and Nelson-Aalen curves were estimated for the overall data and compared with log-rank test. The significance level for all statistical tests was set to 0.05. However, due to small sample size at hand, we also reported marginally significant effects with *p* < 0.1. Stata/MP 17 was used for computations..

## 3. Results

Between May 2018 and June2022, 38 patients with advanced ER+/HER2- BC received palliative metronomic FulVEC chemo-endocrine treatments. The median age of patients at the FulVEC initiation was 46 years (30–80), and the median follow-up was 22 months. Most patients had bone (82%) and/or liver (66%) metastases, with only 8% of patients presenting bone-only disease. Almost half of the analyzed patients (47%) failed previous endocrine treatments with CDK4/6 inhibitors administered in the first line (27.7%) or ≥ second line of therapy (19.3%). Most patients failed previous endocrine treatments with fulvestrant (53%) and experienced progression during chemotherapy, including at least one cytotoxic agent used in the FulVEC regimen (48.7%). Detailed patients’ characteristics are included in Table 1.

### 3.1. Efficacy

Biochemical response in the evaluable group (n:38) was observed in 19 (48.7%) patients, with the complete normalization of CA-15.3 confirmed in 8 patients (20.5%). In general, 30 patients (76.9%) experienced a decline in the serum marker (Figure 1). Patients who had previously received treatment based on any drugs included in the FulVEC regimen had non-significantly lower bRR and bSD compared to individuals not exposed to any FulVEC’s component. The bRR and bSD rates were 70% and 20% in non-pretreated patients and 50% and 28.6% in pretreated patients, respectively (Figure 2A).

Previous treatments with fulvestrant were associated with a higher risk of biochemical progression—30% compared to 11.1% in fulvestrant-naïve patients (Figure 2B).

Biochemical stabilization (any decline in CA-15.3 levels) was observed in a similar percentage of patients irrespectively of previous exposure to any of the cytotoxic drugs used in the FulVEC regimen (73.7% in the pretreated and 89.5% in the non-pretreated population). However, the bRR was more often observed in non-pretreated patients (73.7%) than in patients who received any cytotoxic component of the FulVEC regimen (42.1%) (Figure 2C). There was no significant difference in the biochemical responses with respect to previous exposure to CDK4/6i. However, numerically, the risk of biochemical progression on FulVEC was at least twice as high in CDK4/6i non-pretreated patients than in pretreated patients (Figure 2D).

The median progression-free survival in the entire population was 8.4 months (95%CI 6.5–11.6) with 12- and 24-month PFS rates of 21% and 3%, respectively (Figure 3A). The median overall survival of FulVEC-treated patients was 21.5 (95%CI 16.7–27.6) with 12- and 24-month OS rates of 63% and 21%, respectively (Figure 3B).

There were no significant differences in PFS irrepective of previous exposure to CDK4/6i, fulvestrant, or FulVEC cytotoxic components (Figure 4A–C). However, patients pretreated with CDK4/6i had a higher median PFS of 9.7 months (95% CI, 5.2–11.6) compared to non-pretreated patients (7.3 months—95% CI, 6.0–14.0). Similarly to PFS, previous exposure to CDK4/6i, fulvestrant or FulVEC cytotoxic components had no impact on OS (Figure 4D–F).

### 3.2. Safety

The most frequent type of AE was myelotoxicity, with neutropenia observed in 41% of patients (G3-4 in 25.7%); however, no cases of neutropenic fever were observed (Table 2). Additionally, the most frequent non-hematologic AE was the capecitabine-related hand-foot syndrome (HFS), which occurred in 12.8% of patients (G3-4 in 2.5%). Treatment-related AEs leading to dose reduction in 46% of patients were most often associated with myelotoxicity (80%) or HFS (15%).

The AE-related temporary treatment interruption was required in 23% of patients and was again most often associated with myelotoxicity (70%) or HFS (25%). However, no patient required permanent treatment cessation due to FulVEC-associated toxicity, and all patients could continue the metronomic chemo-endocrine therapy, albeit with some dose reductions (Table 3).

Treatment-related AEs were easily manageable by a precise, step-wise dose-adjustment approach of the FulVEC regimen (Figure 5). Such a precise mode of dose modification is a unique feature of multi-drug chemotherapy regimens, which allow for a minimal decrease in treatment intensity with the direct mitigation of particular treatment-emergent AE.

## 4. Discussion

Even though the concept of metronomic chemotherapy is relatively old, it has not been robustly evaluated in randomized clinical trials. Most available data on the utility of metronomic chemotherapy (MCT) in advanced breast cancer (ABC) come from single-arm, phase II clinical trials or retrospective analyses [10,11,12,13,14,15,16]. The results of a phase III study (METEORA), which were presented recently, demonstrate the superiority of metronomic chemotherapy VEX (VEC) over weekly paclitaxel in terms of PFS and time-to-treatment failure (TTF) endpoints [17]. The METEORA study has thus established the VEC regimen as a safe and active option of MCT for the treatment of advanced BC.

Compared to metronomic chemotherapy, there are even fewer data on the role of a combination of MCT with endocrine treatments. In a single-arm phase II study, 41 ABC patients following ≤1 line of endocrine treatment without previous chemotherapy received metronomic capecitabine combined with fulvestrant [18]. The administration of concurrent chemo-endocrine therapy (CET) induced 24.5% of objective responses and 58.5% of the clinical benefit rate. The median PFS and OS were 15.0 and 28.7 months, respectively, which compares favorably to the outcomes observed in pivotal clinical trials on CDK4/6i combined with fulvestrant. The treatment was well tolerated, with hand-foot syndrome being the most common G3 adverse event observed in 7.3% of patients. A retrospective study by Aurilio et al. evaluated the combination of fulvestrant with metronomic chemotherapy (cyclophosphamide + methotrexate) in 32 heavily pretreated ER+ ABC patients [19]. Concurrent CETs led to one partial response and disease stabilization in 17 patients (53%). Again, the study revealed promising clinical activity of CET with an excellent safety profile. In a phase II study, Rashad et al. evaluated a combination of capecitabine-based chemotherapy with endocrine treatment (letrozole or tamoxifen) as the first-line treatment of ER+ ABC [20]. Concurrent CETs were associated with objective response and clinical benefit rates in 60% and 82.5% of patients, respectively. The median PFS and OS for the general population were 10.0 and 23.3 months, respectively. In patients treated with the capecitabine + letrozole combination, the median PFS and OS were higher, respectively, by 4.0 and 3.0 months than in the capecitabine + tamoxifen combination.

The results of our analysis compare favorably to the above-mentioned studies. FulVEC led to a high rate of clinical benefit (at least stabilization of serum marker with no signs of progression) in most patients who were previously resistant to ET and (in many cases) also to CT. Additionally, unlike other studies conducted in much less pretreated populations, the median PFS of 8.4 months and median OS of 21.5% underscores the activity of FulVEC regimen. Moreover, almost half of the patients treated with FulVEC regimen failed earlier endocrine therapy combined with CDK4/6i, which were not available when the older, above-mentioned studies on CET were conducted. Recent studies on the repeated administration of CDK4/6 inhibitors combined with other endocrine agents applied in patients who failed 1st line ET + CDK4/6i revealed at least some modest activity of such approaches. A phase II study (PACE) was conducted in patients who failed first-line treatment with CDK4/6i+IA. Patients were randomized into three arms: treatment with fulvestrant, fulvestrant + palbociclib, or fulvestrant + palbociclib + avelumab. The repeated administration of CDK4/6i has no impact on fulvestrant’s efficacy with median PFS and OS of 4.6 months and 24.6 months (fulvestrant + palbociclib) and 4.8 months and 27.5 months (fulvestrant), respectively [21]. Another phase II, single-arm study, which evaluated the combination of fulvestrant + palbociclib in patients who failed IA + Palbociclib, demonstrated a short median PFS of 3.7 months [22].

Since phase III studies on the combination of fulvestrant + CDK4/6i in metastatic BC resistant to aromatase inhibitor alone demonstrated a median PFS within the range of 11.2–20.5 months and a median OS of 34.8–46.7 months [23,24,25], the CDK4/6i rechallenge does not represent a viable clinical option. In this context, the results of our analysis provide very promising data on the activity of the FulVEC regimen in metastatic BC patients irrespective of previous therapies (fulvestrant, chemotherapy or CDK4/6i). This phenomenon must be linked to the multidirectional activity of metronomic chemotherapy and the synergism of multi-drug chemotherapy combinations, as well as the simultaneous administration of cytotoxic and hormonal agents.

One of the most important aspects of the FulVEC-based treatment is its safety and the gradual development of treatment-emergent AEs. Despite the fact that some patients experienced G3-4 AEs (mainly neutropenia but without neutropenic fever), none were life-threatening, and all readily subsided after dose adjustments. This safety profile of FulVEC highly resembles the typical toxicity observed in patients treated with CDK4/6i combined with ET. It must be underscored that the construction of FulVEC allows for a uniquely precise and stepwise dose reduction in a particular drug responsible for a specific AE. The dose intensity of FulVEC can be thus be reduced by much smaller steps compared to CDK4/6i (Figure 5), where the relative dose intensity per dose reduction diminished by 20% (palbociclib) or 33% (ribociclib, abemaciclib). Therefore, only a few FulVEC-treated patients required treatment interruption, and none definitely stopped the MCT due to toxicity.

We admit that our analysis of FulVEC regimen has several limitations, and they are mainly related to its retrospective character and the diversity of the studied population. There are also insufficient data on objective tumor responses since regular imaging (outside prospective clinical trials) is rarely clinically meaningful in advanced, treatment-refractory, and often symptomatic patients who demonstrate clinical benefit from a palliative systemic treatment. Additionally, PIK3CA or ESR1 gene mutations, which may have had a significant impact on ET efficacy, have not been determined in the studied population. Nevertheless, the study provides a promising signal on the safety and efficacy of metronomic chemo-endocrine therapy in advanced, often heavily pretreated metastatic BC patients, of whom many exhausted all available treatment options and who represent a real-world population of patients with extremely poor outcomes. Such a population of BC patients desperately awaits novel, active and safe systemic therapies. The robust OS data observed in the FulVEC-treated population stay in conjunction with other endpoints of our analysis (biochemical responses and PFS). Based on the obtained results, a randomized, phase II study is warranted.

## 5. Conclusions

The promising OS data observed in the FulVEC-treated population stay in conjunction with other endpoints of our analysis (biochemical responses and PFS). Based on the obtained results, a randomized, phase II study is warranted.

## Figures and Tables

**Figure 1 jcm-12-01350-f001:**
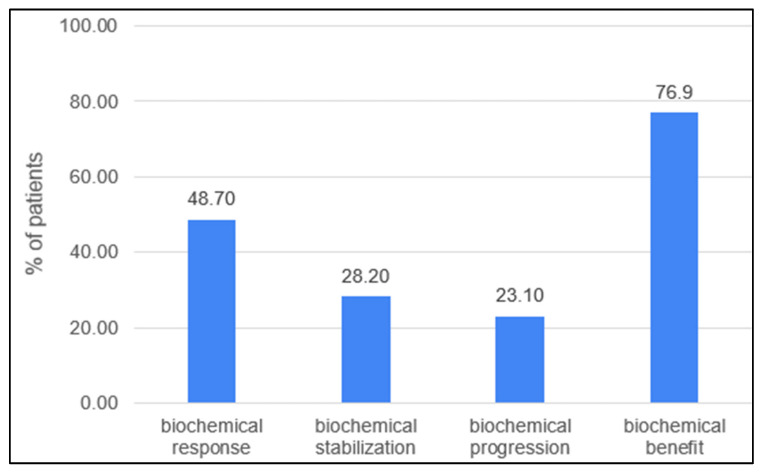
Biochemical efficacy of FulVEC (bRR—≥50% decline in CA-15.3; bSD—1–49% decline in CA-15.3; bPD—any increase in CA-15.3; bBR—any decline in CA-15.3).

**Figure 2 jcm-12-01350-f002:**
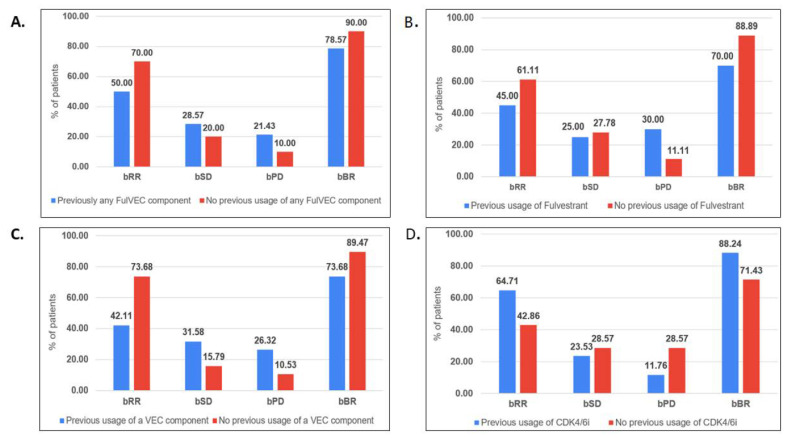
Biochemical efficacy of FulVEC according to previous therapy. (**A**). Previous use of any FulVEC component, (**B**). Previous use of fulvestrant, (**C**). Previous use of any VEC component, (**D**). Previous use of CDK4/6i. (bRR—≥50% decline in CA-15.3; bSD—1–49% decline in CA-15.3; bPD—any increase in CA-15.3; bBR—any decline inCA-15.3). No significant differences (chi-square test) in biochemical response rates in relationships relative to previous treatments were observed in (**A**–**D**).

**Figure 3 jcm-12-01350-f003:**
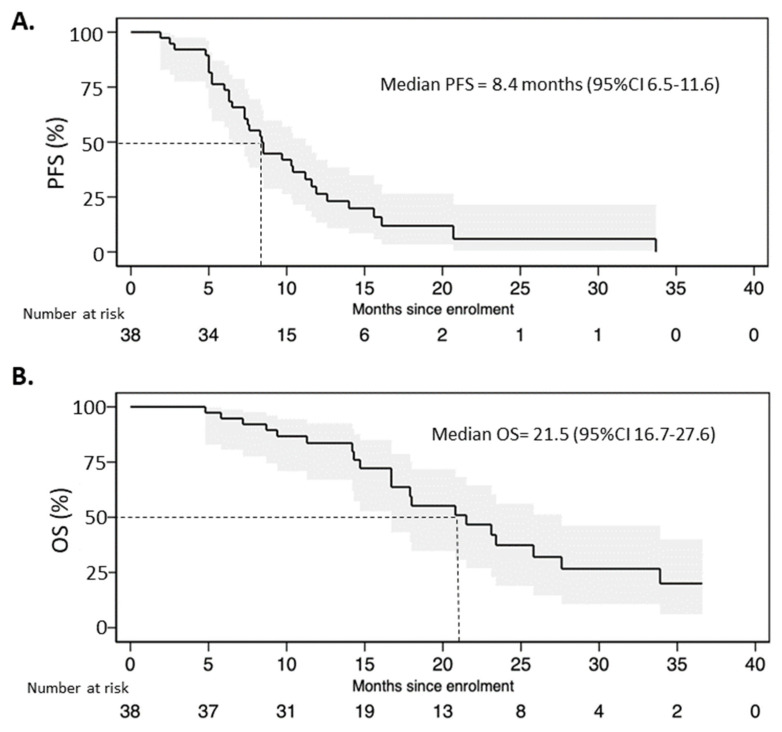
(**A**). Progression-free survival and (**B**) overall survival of patients treated with FulVEC.

**Figure 4 jcm-12-01350-f004:**
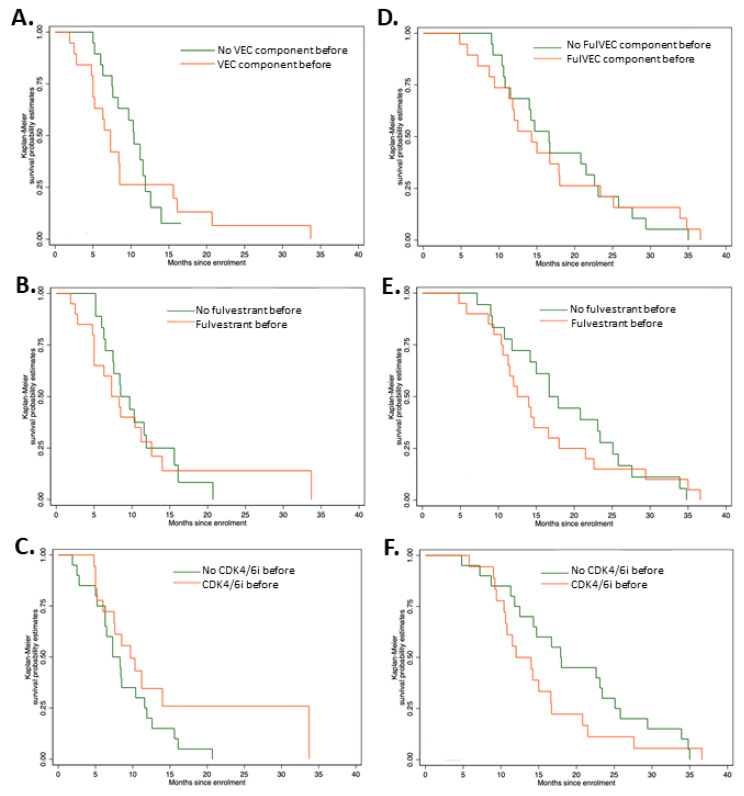
Progression-free survival and overall survival of patients treated with FulVEC. PFS of patients pretreated with (**A**) any cytotoxic component of FulVEC; *p* = 0.502; (**B**) fulvestrant; *p* = 0.739; (**C**) CDK4/6i; *p* = 0.142. Overall survival of patients treated with FulVEC and OS of patients pretreated with (**D**) any cytotoxic component of FulVEC; *p* = 0.284; (**E**) fulvestrant; *p* = 0.898; (**F**) CDK4/6i; *p* = 0.929. All log rank tests for the equality of survival functions (χ^2^).

**Figure 5 jcm-12-01350-f005:**
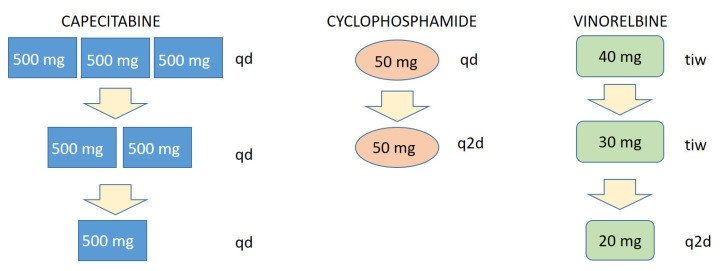
Scheme of a stepwise, particular AE-related dose reduction in FulVEC cytotoxic components. tid—three times a day; bid—two times a day; qd—every day; q2d—every second day; tiw—three times a week.

**Table 1 jcm-12-01350-t001:** Patients’ characteristics.

Parameter	N (%)
Number of patients	38
Median age (range)	46 years (30–80)
Menopausal status	
Premenopausal	15 (39%)
Postmenopausal	24 (61%)
Previous neo/adjuvant chemotherapy	24 (63%)
Previous adjuvant endocrine therapy	23 (61%)
Locoregional relapse	11 (29%)
Disseminated disease	39 (100%)
Bone metastasis	31 (82%)
Bone-only disease	3 (7.7%)
Lung metastasis	13 (34%)
Liver metastasis	25 (66%)
CNS metastasis	3 (8%)
No. of previous systemic treatments	2 (median)
≤2	20 (53%)
3–4	8 (21%)
≥5	10 (26%)
Previously any FulVEC component	30 (77%)
Fulvestrant	20 (53%)
Vinorelbine	11 (29%)
Cyclophosphamide	12 (32%)
Capecitabine	13 (34%)
Previous treatment with CDK4/6i	18 (47%)
1st line	10 (59%)
≥1st line	7 (41%)

**Table 2 jcm-12-01350-t002:** Adverse events in patients treated with the FulVEC regimen.

Adverse Events	Any Grade	G3	G4
Any adverse event	51.3%	38.5%	10.3%
Neutropenia	41.0%	15.4%	10.3%
Anemia	12.8%	5.1%	0%
Thrombocytopenia	2.5%	0%	0%
Fatigue	7.7%	2.6%	0%
Hfs	12.8%	2.5%	0%
Hepatotoxicity	5.1%	2.5%	0%
Abdominal pain	2.5%	2.5%	0%

**Table 3 jcm-12-01350-t003:** Interventions in patients experiencing AE during metronomic chemo-endocrine therapy.

Toxicity-Related Treatment Decisions	% (N)
Dose reduction	46% (18)
Myelotoxicity	80%
H&F syndrome	15%
Other	5%
Temporary treatment interruption	23% (9)
Myelotoxicity	70%
H&F syndrome	25%
Other	5%
Permanent treatment cessation due to AE	0% (0)

## Data Availability

Data is unavailable due to ethical restrictions.

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
