# Peer review of "Metronomic Chemo-Endocrine Therapy (FulVEC) as a Salvage Treatment for Patients with Advanced, Treatment-Refractory ER+/HER2-Breast Cancer—A Retrospective Analysis of Consecutive Patients Data"

_jcm, 2023, doi:10.3390/jcm12041350_

Round 1

Reviewer 1 Report

The authors retrospectively analyzed the efficacy of metronomic chemo-endocrine Therapy (FulVEC) as a salvage treatment in ER-positive/HER2-negative breast cancers.

The manuscript is well documented and suitable for publication in JCM, although some minor corrections are needed.

Line 133. CD4/6 CDK4/6

Line 144. According to Figure 2, P value should be given. Which statistical test was used? Chi-squared test?

Figure 2. Figure 2B is missing while Figure 2C is duplicated.

Line 167. Figure2A Figure 3A

Line 169. Figure 2B Figure 3B

Figures representing KM curves (p6 and p7) may be Figure 3 and Figure 4, respectively. P value should be given.

Following words should be spelled out.

Line 215. ABC

Line 245. IA

Author Response

Dear Reviewer,

thank you for a detailed review of our manuscript entitled “Metronomic chemo-endocrine therapy (FulVEC) as a salvage treatment for patients with advanced, treatment-refractory ER+/HER2- breast cancer – a retrospective analysis of consecutive patients data”

We appreciate your time and expertise in commenting on our study. Below you can find actions we took in response to your comments.

Line 133. CD4/6 → CDK4/6 - corrected

Line 144. According to Figure 2, P value should be given. Which statistical test was used? Chi-squared test? – the information on non-significant differences according to the Chi-square test was added in the text and in the figure legend.

Figure 2. Figure 2B is missing, while Figure 2C is duplicated. – the figure labeling has been corrected

Line 167. Figure → Figure 3A the figure labeling has been corrected

Line 169. Figure 2B → Figure 3B the figure labeling has been corrected

Figures representing KM curves (p6 and p7) may be Figure 3 and Figure 4, respectively. P value should be given. – the p values have been added in the figure legend

The following words should be spelled out.

Line 215. ABC - done

Line 245. IA - done

Reviewer 2 Report

1. The drug administration strategy lacks strong evidence support, just referred to the reference 17, which was a meeting reportand the references was listed incompletely in the manuscript. The basis for dose design and dose adjustment due to ADR was not clearly described.

2. The total number of research subject is 39, which is a small size,and furtherseveral subgroup analyses were performed in such a small size. The sample size in each group was too small to draw valuable conclusions.

3. In terms of pairs comparision in Figure2(like no VEC component before vs VEC component before), the characteristics of the population in each group should be consistent, but the authors did not clarify the patients characteristics in two compared groups.

4. Number at risk should be listed on the survival curves.

5. It was mentioned in the methodology that the time for collecting patients was 2017-2022 (line76), but the time in the results was between 05.2018-06.2022 (line 128).

6. A lot of details have not been checked clearly. Manuscript might not be carefully checked before submission.

1) The marking in the Figure 2 is inconsistent with the annotation under the figure in the manuscript.

2) Capecitabine administration in Figure 4 isqd, which is inconsistent with the description tidin the manuscript .

3) The format of the summary is disorder.

4) In Line92,......was reduced from 50 mg tiw to 30 mg q2. Did the authors mean q2d?

Author Response

Dear Reviewer,

thank you for a detailed review of our manuscript entitled “Metronomic chemo-endocrine therapy (FulVEC) as a salvage treatment for patients with advanced, treatment-refractory ER+/HER2- breast cancer – a retrospective analysis of consecutive patients data”

We appreciate your time and expertise in commenting on our study. Below you can find responses to your comments.

Ad 1 – the drug administration strategy is a unique regimen, which has not been previously published. The concept is based on several earlier studies and analyses demonstrating the feasibility and promising activity of combining metronomic (usually single-drug) chemotherapy with endocrine agents (mostly aromatase inhibitors- AI). We have decided to apply the most intensive and well-studied metronomic chemotherapy regimen (VEC/VEX) which has been evaluated in several phase II trials. Additionally, we have decided to use fulvestrant instead of AI, since fulvestrant demonstrates activity in patients previously exposed to AI or tamoxifen. The detailed dose adjustment strategy was developed by us and was not published before.

Ad 2. We totally agree that the small population size makes any analyses difficult and robust conclusions impossible. However, we assume that the subgroup analyses have a strong hypothesis-generating potential which will be used for the construction (stratification and pre-planned subgroup analyses) of a future phase II, randomized, clinical trial.

Ad. 3. As mentioned before, the retrospective nature of our analysis makes the inter-patient comparisons difficult, especially when there is a huge diversity of subgroups. The comparative analysis of patients according to the previous use of VEC components is impossible since it included patients who could previously receive single-drug therapy based on capecitabine, vinorelbine, or cyclophosphamide, could have received two-drug combinations of thereof, or all above-mentioned agents sequentially. According to your suggestion, we checked if there are any meaningful differences between these subgroups according to the duration of previous treatment, and lines of previous therapy but none were found.

Ad 4. “Number at risk” should be listed on the survival curves.

Ad 5. Thank you for noticing this error. It has been corrected.

Ad 6. 

  1. the marking in Figure 2 is corrected to 2B
  2. the capecitabine administration in Figure 4 is qd (3x 500 mg) à qd (2x500 mg) à qd (1x500 mg) – it is a graphic presentation of the dosage-adaptation scheme.
  3. The format of the summary is disorder.
  4. Line 92 – has been corrected to q2d

Reviewer 3 Report

In this manuscript, Buda-Nowak et al. performed a retrospective study to demonstrate the efficacy and safety of FulVEC treatment on a cohort of advanced breast cancer patients. This study demonstrated if previous treatment with FulVEC, fulvestrant, VEC component, or CDK4/6i affected FulVEC efficacy. Although the results were negative, it provided useful information for developing new treatment for advanced breast cancer patients, especially failed to the endocrine therapy or chemotherapy. However, several questions should be addressed.

1. This is a small cohort. Since it is a retrospective study, it might be fruitless to ask for more patients. But the authors should think about to include control group (no FulVEC treatment) if possible, to better understand if FulVEC treatment could improve patient’s outcome.

2. The previous similar studies (references 18-25) should be included in the introduction section, rather than in the discussion.

3. There are 3 figures labeled as figure 2.  Please label them correctly in the figure and main text.

4. Line 65-66, the author mentioned “many biological similarities between CDK4/6i and metronomic chemotherapy,”. But the figure 2 on page 5 shows the previous usages of VEC component and CDK4/6i have opposite effects on Biochemical efficacy of FulVEC. How to explain?

Author Response

Dear Reviewer,

thank you for a detailed review of our manuscript entitled “Metronomic chemo-endocrine therapy (FulVEC) as a salvage treatment for patients with advanced, treatment-refractory ER+/HER2- breast cancer – a retrospective analysis of consecutive patients data”

We appreciate your time and expertise in commenting on our study. Below you can find responses to your comments.

Ad. 1. We confirm that the retrospective nature of the study precludes the qualification of further patients. However, this analysis provides hypothesis-generating data which will be used in a forthcoming randomized, phase II clinical, which will allow for a direct comparison of patients treated with standard therapies and the experimental (FulVEC) regimen.

  1. We have included the references in the introduction but discussed them thoroughly in the discussion

  1. The numbers have been corrected.

  1. There were no significant differences in the efficacy of FulVEC irrespectively of previous treatment (fulvestrant or vinorelbine/cyclophosphamide/capecitabine or CDK4/6i). Such information is included in figure 2.

Round 2

Reviewer 2 Report

The sequence number in the abstract starts from (2), and (1) is missing. And the word “result” in the abstract is missing. Number at risk is still not listed in the survival curve. There are  still two C  in Figure 2,but B missing.The usage of capecitabine was still described as qd, while it was describes as tid in manuscript. 

Author Response

The response to the reviewer's comments. 

Thank you for the additional comments 

  • Abstract - The "(2)" before Methods was removed
  • Abstract - The "Results" has been added
  • Figure 2 has number at risks listed 
  • Figure 2 does not have C graph, in Figure 3 all descriptions are correct
  • Since the Reviewer could not interpret figure 4 clearly, we have modified it to avoid any confusion. 

Reviewer 3 Report

Two figures are labeled as figure 4. Please correct.

Author Response

Thank you very much. We have corrected the manuscript with all the issues addressed (marked yellow).